# MAKING MATHEMATICAL REASONING ADAPTIVE

## ABSTRACT

Mathematical reasoning is a primary indicator of large language models (LLMs) intelligence. However, existing LLMs exhibit failures of robustness and generalization. This paper attributes these deficiencies to spurious reasoning—i.e., producing answers from superficial features. To address this challenge, we propose the AdaR framework to enable adaptive reasoning, wherein models rely on problem-solving logic to produce answers. AdaR synthesizes logically equivalent queries by varying variable values, and trains models with Reinforcement Learning with Verifiable Rewards (RLVR) on these data to penalize spurious logic while encouraging adaptive logic. To improve data quality, we extract the problem-solving logic from the original query and generate the corresponding answer by code execution and then apply sanity check. Experimental results demonstrate that AdaR improves robustness and generalization, achieving substantial improvement in mathematical reasoning while maintaining high data efficiency. Analysis indicates that data synthesis and RLVR function in a coordinated manner to enable adaptive reasoning in LLMs. Subsequent analyses derive key design insights into the effect of critical factors and the applicability to instruct LLMs.

## 1 INTRODUCTION

Large Language Models (LLMs) have demonstrated strong performance across various reasoning tasks (Wei et al., 2022a; Huang & Chang, 2023). Among these, mathematical reasoning serves as a crucial cognitive skill that supports problem-solving across tasks (Huang & Chang, 2023). Beyond early direct inference attempts (Liu et al., 2021; Brown et al., 2020) (the *black arrow* in Figure 1), Chain-of-Thought (CoT) has been recognized as an effective approach to enhance mathematical reasoning (Wei et al., 2022b), as it breaks down complex problems into manageable steps and offers interpretability by making the reasoning process transparent and trustworthy (Chu et al., 2023).

However, existing mathematical LLMs still exhibit failures at two levels: (i) robustness on in-domain tasks (Mirzadeh et al., 2024); and (ii) generalization on out-of-domain tasks (Jahin et al., 2025). We show that[1] these deficiencies arise from **spurious reasoning** (the *red arrow* in Figure 1), the process by which LLMs derive gold answer $y$ from superficial features but not the correct problem-solving logic $L$, therefore producing reasoning trace $z$ (i.e. CoT) that bear negligible causal connection to $y$. Consequently, even when the underlying problem-solving logic $L$ remains unchanged, models relying on spurious reasoning fail to adapt to numerical changes of values in the variable set $x$ and exhibit instability in performance. Meanwhile, spurious reasoning, which does not rely on $L$, is non-compositional along causal relations and therefore make models generalize ineffectively.

We argue that an ideal reasoning process, i.e. **adaptive reasoning**, should rely on correct problem solving logic, enabling LLMs to adapt to varying values of $x$ and to exhibit stronger generalization. This property exemplifies algebraic thinking (Kieran, 2004); accordingly the process could be as follows (the *green arrow* in Figure 1): The model is required to decompose a query $q$ into a template $T$ by algebraic abstracting the variables, with $T$ serving as a key feature; concurrently, the abstraction produces a specific mapping that is recorded as $x$. Meanwhile, the LLM must model the underlying $L$ as a function over $x$, conditioned on $T$. By applying the decomposed $x$ into $L_T$, the LLM can generate text form of $L_T(x)$, and finally obtain the answer $y$.

While Reinforcement Learning with Verifiable Rewards (RLVR) aims to enhance generalization without human supervision (Lambert et al., 2025; Guo et al., 2025a), it relies exclusively on outcome

---

[1]Please refer to Section 4.1.

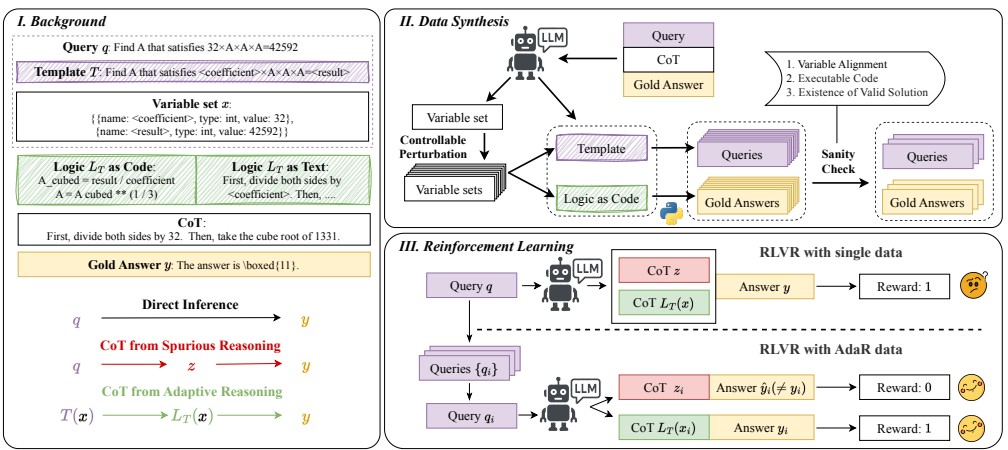

Figure 1: Subfigure I presents the composition of the math data and the modeling of three reasoning processes. Subfigure II illustrates how we get the Query-Answer pairs by controllably perturbing variable values while preserving problem-solving logic and sanity. Subfigure III illustrates that synthetic data, when leveraged via RLVR, elicits adaptive reasoning by comparing rewards from responses to perturbed queries.

correctness as the reward signal, regardless of whether the response was derived through spurious or adaptive reasoning. Consequently, this outcome-centric feedback mechanism may inadvertently strengthen existing spurious reasoning.

Inspired by evidence that humans induce problem-solving logic through comparison to acquire **Ada**ptive **R**easoning (Gerstenberg et al., 2015; Ullman, 2015), we propose the **AdaR** framework, including a data synthesis part and a model training part. AdaR synthesizes diverse data by keeping the problem-solving logic unchanged and perturbing the values in the variable set. Two challenges must be solved when performing data synthesis: preserving sanity while perturbing, and obtaining the gold answer without human annotations. To address these challenges, we decompose the overall complex task into the following manageable, verifiable sub-tasks. As shown in subfigure II of Figure 1, we first prompt an open-source LLM, generating a template corresponding to the query, a problem-solving logic rendered as code (e.g. a Python program), and a variable set. Subsequently, we *controllably perturb* the values in identified variable set to predefined magnitudes and types. Perturbed variable sets are then used to instantiate the template to generate new queries, and are also provided as input to the code, which is executed to produce gold answers. Furthermore, we introduced a *sanity check* to filter invalid instances.

AdaR then trains the model with RLVR to improve the adaptive reasoning. Notably, unlike in the single data situation, where determining whether a response derives from spurious reasoning or adaptive reasoning is infeasible, the correctness of outcomes on these synthetic data provides a reliable signal for inferring where their responses derive. In detail, responses that rely on spurious reasoning are more likely to produce incorrect answers on the perturbed synthetic data and are consequently penalized in RLVR, thereby pushing the model to explore the adaptive problem-solving logic (as shown in subfigure III of Figure 1).

Extensive experiments across in-domain robust tasks and out-of-domain tasks demonstrate that AdaR achieves great gains (+8.50 points on average), with only 9K synthetic data, thereby demonstrating that our approach enhances the model's robustness and generalization. Further analysis indicates that: (i) all components of AdaR contribute to its performance, with the combination of synthetic data and RLVR being a vital element we wish to highlight; (ii) Evidence of improved algebraic thinking and heightened influence on logical order demonstrates that AdaR enables adaptive reasoning. (iii) the magnitude of perturbation balances the exploration scope and data quality; (iv) scaling variable values greatly facilitates adaptive reasoning compared to scaling query template; (v) for a given template, instances featuring perturbed variable values should be presented to the LLM for comparison; (vi) AdaR is applicable to instruct model.

## 2 METHOD

In this section, we first present methods for synthesizing data that ensure controllability and sanity throughout the generation process. Following this, we introduce our training strategy, which is employed to more effectively integrate with synthetic data, preventing the model from learning spurious reasoning, thereby facilitating adaptive reasoning.

### 2.1 DATA SYNTHESIS WITH EXECUTABLE CODE AND VERIFIABLE ANSWERS

A straightforward method for perturbing values in queries and obtaining correct answers without human annotation is to prompt a LLM (Wang et al., 2025b). However, our preliminary experiments indicate that this approach offers neither explicit control over the magnitudes and types of perturbations nor any guarantee of answer correctness. To synthesize the desired query-answer pairs, AdaR constrains the model to identify only the metadata within the data and to translate a CoT into code; controllable perturbations are then applied externally, and code execution is used to ensure answer correctness. The AdaR framework accordingly decomposes it into a sequence of manageable, verifiable sub-tasks as follows.

**Convert Logic in Text to Logic in Code.** Executable problem-solving code can serve as a problem-solving logic provided its correctness is guaranteed. Crucially, such code generalizes effectively: by substituting the input variable set, it can solve any perturbed query of the original query, for its output can serve as reliable gold answer. When CoT and gold answer are provided, problem-solving code generation becomes a straightforward process of translating the textual logic (i.e. CoT) into executable code, thereby reducing the model's reasoning burden and enabling direct verification of correctness against the gold answer. Building on these insights, we provide a query, the corresponding CoT and gold answer to an open-source LLM to synthesize the associated problem-solving code for producing gold answers of subsequent perturbed queries. Because code generation requires abstracting the concrete numerical values in the query into a mapping to the input variables, we further instruct the model to produce a query template as a byproduct by applying this mapping into the original query. Appendix A.2 details the prompt employed.

**Controllable Perturbation.** By comparing the query with the generated template, we construct a variable set that records each variable's name, numeric type and its value as it appears in the original query. This variable set, enriched with metadata, enables direct numeric perturbations at a predefined magnitude while ensuring numerical validity, thereby avoiding the unpredictability of perturbations performed through an LLM. Specifically, we then apply independent perturbations by sampling each variable's value within a range of $\pm\alpha\%$ of its original value. The parameter $\alpha$ allows the control of the perturbation magnitude. To ensure numerical validity, both the numeric type and the sign of every variable must remain unchanged before and after perturbation. Formally, for the $i$-th variable with original value $x_0^i$, we draw:

$$x^i = x_0^i \times (1 + \Delta_i), \quad \text{where } \Delta_i \sim \text{Uniform}(-\alpha\%, \alpha\%)$$
$$\text{s.t.} \quad \text{type}(x^i) = \text{type}(x_0^i), \; \text{sign}(x^i) = \text{sign}(x_0^i) \tag{1}$$

**Sanity Check.** After applying perturbations, we instantiate the template with each variable set to obtain perturbed queries, and execute problem-solving code with the same variable sets as input to generate the corresponding gold answers. Although providing the CoT and answer has reduced task difficulty, query template and problem-solving code generated by the LLM remain uncertain and may contain errors. Furthermore, a meaningful query imposes inter-variable constraints on its variables; independent perturbations can violate these constraints and thereby introduce errors. To ensure that the perturbed data remain well-posed, we conduct a *sanity check* along the following aspects:

- **Variable Alignment (VA).** We compare variables referenced in each query template with those used in each problem-solving code. Any mismatch indicates potential errors (e.g. hallucination) in the LLM's output.
- **Executable Code (EC).** Since the problem-solving code is used to derive the gold answer, executability is a critical requirement: (i) the code runs without runtime errors; and (ii) providing the original variable set $x_0$ as input reproduces the gold answer $y_0$.

- **Existence of Valid Solution (EVS).** As perturbations do not incorporate inter-variable constraints, perturbed queries may be invalid in realistic scenarios (e.g., selecting 20 items from a set of 10). To provide evidence that a valid solution exists, we perform cross-validation by comparing the gold answer generated by the code with the output of a mathematical LLM under perturbed query input. Nevertheless, because post-training can introduce spurious reasoning, the model may produce a incorrect answer when conditioned solely on the perturbed query. To handle this, we supply the corresponding problem-solving code as a hint, which has been verified by EC, enabling the model to optionally ground its reasoning on the logic underlying the code.

If a perturbed instance fail to pass the *sanity check*, we reattempt *controllable perturbation*. If the number of attempts exceeds $\tau$ times, we conclude this instance likely involves complex inter-variable constraints and then discard it from synthesis.

**Queries Paraphrasing.** We adopt paraphrasing (Yu et al., 2023) as a data augmentation strategy to increase the diversity of the query template $T$. This approach complements the *controllable perturbation* introduced earlier to increase the diversity of the variable set $x$. The impact of these scalable dimensions on model performance is further explored in Section 3.3 and Section 4.3.

**Remark.** *Our synthetic data contains no CoT, it consists solely of the query and the corresponding gold answer. As will be described in the next subsection, CoTs generated through either spurious reasoning or adaptive reasoning can be sampled from the target LLM.*

## 2.2 TRAINING STRATEGY

Supervised fine-tuning (SFT) is a mainstream post-training strategy for eliciting step-by-step reasoning. Let $\pi_\theta$ denote a policy over responses given queries, parameterized by $\theta$. Given a dataset $\mathcal{D}$ of query–response pairs $(q, r)$, SFT minimizes the negative log-likelihood objective

$$\mathcal{L}_{\text{SFT}}(\theta) = -\mathbb{E}_{(q,r)\sim\mathcal{D}}\big[\log \pi_\theta(r \mid q)\big], \tag{2}$$

but it makes models prone to memorizing provided CoTs rather than developing adaptive reasoning (Chu et al., 2025), making it fall into a local optimum Kang et al. (2025). Rejection sampling Fine-Tuning (RFT) constructs an SFT dataset by sampling from the model to be trained and retaining high-scoring outputs. Although it avoids reliance on the responses distilled from powerful LLMs exist in the training data and is more robust than vanilla SFT (Yuan et al., 2023), it remains susceptible to memorization of superficial features, leading to a generation problem.

RLVR has recently been widely adopted to improve generalization (Guo et al., 2025a). Given a query $q$, the model samples $r \sim \pi_\theta(\cdot \mid q)$ and a verifier $v(q, r) \in [0, 1]$ evaluates it. RLVR maximizes

$$J(\theta) = \mathbb{E}_{q\sim\mathcal{D}, r\sim\pi(\cdot\mid q)}[v(q, r)] \tag{3}$$

In mathematical reasoning, $v$ typically checks whether the predicted answer exactly matches the gold answer. However, the outcome produced via either spurious reasoning or adaptive reasoning is indistinguishable under this reward, which can inadvertently reinforce the spurious reasoning.

To address this, we combine RLVR with our synthetic data. Whereas SFT/RFT tend to memorize provided CoTs, RLVR's reward-driven exploration weakens this tendency and enables the model to learn from comparison among rewards obtained by solving perturbed queries using different reasoning process. Specifically, when given the original query $q$, it's hard to determine where the gold answer $y$ is derived. However, when the model is evaluated on perturbed queries $q_i$, reliance on CoT from spurious reasoning $z_i$ is more likely to yield incorrect outcome $\hat{y}_i$, whereas reliance on CoT from adaptive reasoning $L_T(x_i)$ is more likely to yield correct outcome $y_i$. All perturbed queries $\{q_i\}$ are placed into the same batch. Then, each reasoning process is more likely to receive appropriate feedback, thereby promoting adaptive reasoning, as illustrated in subfigure III of Figure 1.

## 3 EXPERIMENT

### 3.1 EXPERIMENTAL SETUP

**Data synthesis.** We use the Qwen2.5-72B-Instruct (Yang et al., 2024) as the open-source LLM to generate the query templates and the problem-solving codes. We select $9K$ instances from ORCA-

Table 1: Performance comparison across In-Domain and Out-of-Domain mathematical benchmarks. The datasets "main", "p1", and "p2" are from GSM-SYM. Best results are highlighted in bold.

| Method | In-Domain | | | | | Out-of-Domain | | | | AVG |
|---|---|---|---|---|---|---|---|---|---|---|
| | GSM8K | ORCA-AdaR | main | p1 | p2 | MATH | College | Theorem | AIME | |
| **Qwen2.5-MATH (7B MATH-Specialized Base Model)** | | | | | | | | | | |
| Initial SFT | 81.43 | 77.12 | 74.26 | 64.48 | 52.44 | 42.84 | 28.55 | 14.88 | 4.27 | 48.92 |
| Standard-RLVR | 86.96 | 81.80 | 81.34 | 74.32 | 64.68 | 61.74 | 29.48 | 20.38 | 10.94 | 56.85 |
| MetaMATH | 84.61 | 82.44 | 79.78 | 71.24 | 63.28 | 66.92 | 39.25 | 26.13 | 9.38 | 58.11 |
| MathGenie | 87.64 | 84.96 | 80.40 | 72.60 | 63.56 | 55.94 | 21.11 | 20.05 | 10.31 | 55.17 |
| AdaR | 91.81 | 86.08 | 89.72 | 82.74 | 73.24 | 75.90 | 48.62 | 37.13 | 14.27 | **66.61** |
| **DeepSeekMath (7B MATH-Specialized Base Model)** | | | | | | | | | | |
| Initial SFT | 70.28 | 65.48 | 61.70 | 50.26 | 29.36 | 28.44 | 22.32 | 11.50 | 0.21 | 37.73 |
| Standard-RLVR | 81.43 | 72.56 | 74.76 | 64.34 | 40.00 | 39.26 | 21.61 | 20.38 | 0.73 | 46.12 |
| MetaMATH | 80.89 | 70.28 | 75.14 | 64.54 | 39.56 | 38.02 | 19.13 | 22.63 | 0.21 | 45.60 |
| MathGenie | 80.13 | 73.44 | 74.18 | 64.02 | 40.04 | 39.16 | 22.39 | 23.75 | 3.33 | 46.72 |
| AdaR | 81.43 | 75.44 | 76.56 | 64.94 | 42.00 | 41.18 | 24.23 | 25.25 | 3.33 | **48.26** |
| **Llama3 (8B General Base Model)** | | | | | | | | | | |
| Initial SFT | 67.17 | 57.84 | 59.98 | 47.30 | 24.92 | 18.20 | 9.33 | 8.13 | 0.00 | 32.54 |
| Standard-RLVR | 73.09 | 55.68 | 67.06 | 51.58 | 25.16 | 19.02 | 9.01 | 9.00 | 0.00 | 34.40 |
| MetaMATH | 74.52 | 59.96 | 70.74 | 56.58 | 26.72 | 19.68 | 9.04 | 9.38 | 0.00 | 36.29 |
| MathGenie | 72.47 | 58.90 | 67.26 | 53.72 | 25.38 | 18.30 | 8.92 | 9.00 | 0.00 | 34.88 |
| AdaR | 77.77 | 61.52 | 73.96 | 56.90 | 30.28 | 20.48 | 9.62 | 9.63 | 0.00 | **37.80** |

MATH (Mitra et al., 2024) as seed data for data synthesis. Using these seed data, AdaR synthesizes instances with a predefined magnitude $\alpha = 500$ and a maximum of $\tau = 50$ attempts. For each data in seed data, we select one corresponding synthetic data to construct the "ORCA-AdaR-train", which contains $9K$ instances. We select another $2.5K$ instances to form "ORCA-AdaR-test", ensuring no overlap with "ORCA-AdaR-train". The details are shown in Appendix A.3.1.

**Models.** To prove the adaptability of our framework, we conduct experiments on two categories of base models: math specialized base LLM, specifically Qwen2.5-Math-7B (Yang et al., 2024) and DeepSeekMath-7B (Shao et al., 2024), and 8B general base LLM, specifically LLaMA3-8B (Grattafiori et al., 2024). In analysis, we further explore the deployment of AdaR on Instruct Model.

**Evaluation.** For a comprehensive evaluation of mathematical reasoning, we adopt 7 benchmarks covering both in-domain and out-of-domain evaluation. Specifically, we evaluate in-domain mathematical competence using GSM8K (Cobbe et al., 2021) and evaluate in-domain robustness using ORCA-AdaR-test (ORCA-AdaR) and GSM-SYM (main/p1/p2) (Mirzadeh et al., 2024). Within GSM-SYM, the main, p1, and p2 subsets exhibit increasing difficulty. For out-of-domain evaluation, we use MATH (Hendrycks et al., 2021), CollegeMath (Tang et al., 2024), TheoremQA (Chen et al., 2023), and American Invitational Mathematics Examination (AIME) problems for 2025 to evaluate generalization. Results are reported as pass@1 for all datasets except AIME 2025, for which, in accordance with (Yu et al., 2025), we report avg@32. Further details about the evaluation setup and benchmarks are provided in the Appendix A.3.3.

**Baseline.** We primarily compare our framework, AdaR, against the following baselines: (1) The base models undergo SFT using 9K seed data—this same data is also leveraged for data synthesis. These models are referred to as the "Initial SFT" models. (2) In "Standard-RLVR" setting, the "Initial SFT" is subsequently trained with RLVR on another 9K subset of ORCA-MATH, with its templates being disjoint from the training set used for "Initial SFT". (3) We also include other synthesis methods, such as MetaMATH (Yu et al., 2023), which improves query diversity through paraphrasing and self-verification; MathGenie (Lu et al., 2024), which generate questions by applying back-translation to paraphrased responses. For all RLVR training procedures, we adopt DAPO (Yu et al., 2025) for its faster convergence and the elimination of the need for a value model, which together reduce training time and computational resource requirements. Additional details of the training setup are provided in the Appendix A.3.2.

Table 2: Ablation Study.

| Method | Strategy | Sanity Check | | | Paraphrase | In-Domain | Out-of-Domain | | | |
|--------|----------|------|------|------|------------|-----------|-------|---------|----------|------|
| | | VA | EC | EVS | | | MATH | College | Theorem | AIME |
| Initial SFT | - | - | - | - | - | 69.95 | 42.84 | 28.55 | 14.88 | 4.27 |
| AdaR | RFT | ✓ | ✓ | ✓ | ✓ | 72.91 | 45.20 | 30.41 | 16.13 | 3.54 |
| | RLVR | ✗ | ✓ | ✓ | ✓ | 82.32 | 71.58 | 44.14 | 32.50 | 7.92 |
| | RLVR | ✓ | ✗ | ✓ | ✓ | - | - | - | - | - |
| | RLVR | ✓ | ✓ | ✗ | ✓ | 83.57 | 72.94 | 46.56 | 33.25 | 7.92 |
| | RLVR | ✓ | ✓ | ✓ | ✗ | 84.43 | 75.28 | 48.22 | 34.88 | 6.88 |
| | RLVR | ✓ | ✓ | ✓ | ✓ | **84.72** | **75.90** | **48.62** | **37.13** | **14.27** |

## 3.2 MAIN RESULTS

The main results are summarized in Table 1. We highlight the following three observations:

**Observation 1: Our method with a small amount of synthetic data yields substantial performance gains.** Using only 9K synthetic data, our method surpasses other methods across all base models. Compared to other synthetic data methods, we improve over MetaMATH by 8.50 points and over MathGenie by 11.44 points on average.

**Observation 2: Our method significantly enhances model robustness and generalization.** The enhancement manifests across three levels: (i) perturbed variable values in seen queries during training (ORCA-AdaR-test); (ii) perturbed variable values in unseen queries during training (GSM-SYM main/p1/p2, where main, p1, and p2 represent increasing levels of difficulty); (iii) out-of-domain data. Across all base models, we observe gains on the first two levels compared to the suboptimal MetaMATH (+4.66 / +4.75 points). At a finer level of granularity, we observe that "Initial SFT" rarely produces correct answers across all perturbations of variable values within a specific query template. In contrast, AdaR significantly improves this capability. Besides, on the third level we obtain larger improvements: when using Qwen2.5-MATH-7B as the base model, the average gain reaches +8.56 points. These results indicate that the model generates CoTs from adaptive reasoning rather than spurious reasoning, which better supports robustness and generalization.

**Observation 3: The effectiveness of our method correlates with the mathematical reasoning ability of the base model.** Inspection the "Initial-SFT" results shows the ordering Qwen2.5-MATH > DeepSeekMath > LLama3. Training these models produces corresponding improvements of +17.69, +10.53, +5.26 points, respectively. It implies sufficient mathematics-related knowledge is a necessary condition for sampling both positive and negative responses, thereby facilitating the learning of adaptive reasoning. The result also suggests that AdaR is complementary to the mathematical pre-training with a large amount of real or synthetic mathematical data.

## 3.3 ABLATION STUDY

The AdaR framework employs techniques to ensure the model can master adaptive reasoning from controllable and diverse synthetic data: (i) deploying the RLVR training strategy; (ii) introducing the *sanity check*, comprising Variable alignment (VA), Executable Code (EC), and Existence of Valid Solution (EVS); (iii) incorporating the paraphrase to further enhance the diversity of templates.

As shown in Table 2, each technique within the AdaR framework is essential to the final performance. Notably, without EC, directly applying subsequent EVS retains only $0.2\%$ data, which is insufficient to support training; consequently, no result is reported for this configuration. Although this indicates that EVS functionally contains EC, the rule-based EC is more computationally efficient than the model-based SE, by $218\times$ (as shown in Appendix A.5), and thus remains essential. Besides, the result proves that paraphrase is complementary to perturbation of variable values, since it enhances the ability to understand different templates of specific problem-solving logic.

To further examine the importance of RLVR, we evaluate reasoning ability by measuring the accuracy of problem-solving code generation and computational ability by measuring the accuracy when the model itself is instructed to execute the code (the full prompt is provided in Appendix A.2). As summarized in the Table 3, relative to "Initial-SFT", "AdaR-RFT" improves computation while

Table 3: Evaluation of reasoning and computation ability on ORCA-AdaR-test.

| Method | ORCA-AdaR-test | Reasoning | Computation |
|---|---|---|---|
| Initial-SFT | 77.12 | 75.24 | 54.92 |
| Standard-RLVR | 81.80 | 69.52 | 68.52 |
| AdaR-RFT | 82.32 | 72.36 | 75.36 |
| AdaR | **86.08** | **80.96** | **90.76** |

degrading reasoning, whereas "AdaR" with RLVR yields substantial improvements in both reasoning and computation ability. It is consistent with Chu et al. (2025): RFT's objective tends to memorize superficial features, thereby exacerbating overfitting to computation patterns rather than the corresponding problem-solving logic, while RLVR improves both abilities by encouraging the maximization of the exploration reward.

# 4 ANALYSIS

In this subsection, we address the following Research Questions (RQs) regarding AdaR: (1) Does training with AdaR enable the model to master adaptive reasoning? (2) To what extent does diversity in the computational difficulty levels of synthetic queries affect the model's performance? (3) To what extent does the scaling along different dimensions (query template $T$, variable set $x$, and problem-solving logic $L$) influence the model's performance respectively? (4) During the RLVR stage, when employing synthetic data whose templates are unseen by the target LLM, does AdaR continue to exhibit strong performance? (5) Is AdaR applicable to Instruct models? To answer these RQs, we conduct the following analysis using Qwen2.5-MATH-7B as the base model. Because the AIME evaluation metric differs from those of the other out-of-domain benchmarks, we exclude AIME from the analysis for better reporting.

## 4.1 AdaR ENABLES ADAPTIVE REASONING

**Enhancing algebraic thinking.** We further examine model outputs and observe that, despite no code data being provided during training, the proportion of CoTs containing structural code snippets (e.g. `... B + H = 157 - (23 + 41). Substituting the value of H into the equation, we get ...`) increases from 55% to 90% after training with AdaR (please refer to Appendix A.4 for more details), indicating the emergence of algebraic thinking, treating unknown and known variables on equal footing and solving queries via variable calculation. We then likewise evaluate reasoning ability by measuring the accuracy of problem-solving code generation which engages algebraic thinking. As shown in Table 3, although RLVR is employed in both, training on standard data (Standard-RLVR) decreases accuracy, whereas training on our synthetic data with queries which are semantically similar but instantiate different variables improves accuracy.

**Enhancing the influence to logical order.** Spurious reasoning that relies on superficial features can derive answers without adhering to a correct logical order. By contrast, adaptive reasoning requires strict adherence to the correct logical order to derive answers step by step. To quantify this effect, we introduce the metric, Influence to Logical Order (ILO), which measures the relative change rate in the perplexity (PPL) of correct answers when the sentence order of the corresponding CoT is shuffled.

$$\text{ILO} = \frac{1}{n} \sum_{i=1}^{n} \frac{\left| \text{PPL}(y \mid q, z) - \text{PPL}(y \mid q, R^i(z)) \right|}{\text{PPL}(y \mid q, z)}, \tag{4}$$

where $q$ denotes the query, $z$ denotes the CoT, $y$ denotes the answer, $R$ denotes a random permutation function operating at the sentence level, and $n$ denotes the number of random permutations (set to 5 to keep variance small). We find that the ILO of CoTs unable to generate correct answers across all perturbations of variable values within a specific query template is significantly lower than that of CoTs capable of producing correct answers across all such perturbations (114.24% vs. 221.87%). This finding demonstrates that the ILO metric can effectively distinguish between spurious reasoning and adaptive reasoning. Based on ILO, we show that AdaR significantly improves the adaptive reasoning capability over "Initiate SFT" (150.49% vs. 119.22%).

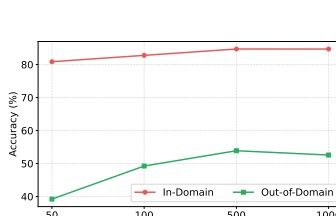 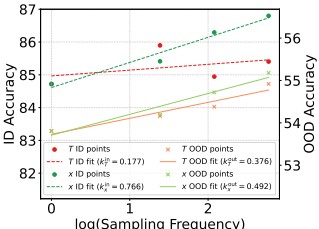 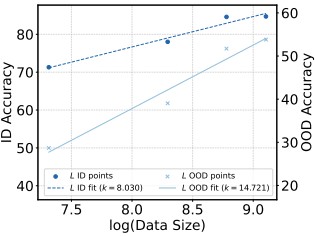

Figure 2: Influence of perturbation magnitude.

Figure 3: Performance of scaling $x$ or $T$.

Figure 4: Performance of scaling $L$.

## 4.2 EFFECT OF PERTURBATION MAGNITUDE

We investigate the influence of perturbation magnitude and present the results in Figure 2. We observe that the performance increases with larger perturbation in variable values up to a point and then exhibits a slight decline. Larger perturbation magnitude enables the model to learn previously unseen numerical calculations, and thereby improves performance. However, when the generated queries contain a excessively high proportion of invalid data (e.g. selecting 2 items form a set of 10 is the original query, the maximum $\alpha$ can be set is $500$) before to the *sanity check* (refer to Section 2.1), it leads more noise that ultimately degrades model performance.

## 4.3 EFFECT OF SCALING DIFFERENT DIMENSIONS

In AdaR, we modulate sampling frequency to scale the query template $T$ and values in the variable set $x$. Accordingly, we evaluate performance trends when scaling $x$ or $T$ and show the result in Figure 3. We observe steady improvements in model performance with increasing sampling frequency in both in-domain (ID) and out-of-domain (OOD) settings. To characterize the marginal return with scale, we fit the scaling curves with a log-linear approximation. The fitted coefficients satisfy $k_T < k_x$ throughout, indicating that scaling along the variable set dimension $x$ yields greater marginal returns than scaling along the query template dimension $T$. This is mainly because perturbing $T$ facilitates query comprehension, whereas perturbing $x$ is the essential driver of adaptive reasoning. For a more comprehensive analysis, we also evaluate the effect of scaling along the problem-solving logic by increasing the data size of the seed dataset used for synthesis. The results are shown in Figure 4. Overall, these three dimensions are complementary, but the cost of scaling $L$ is substantially higher; we therefore leave this point to future work.

## 4.4 EFFECT OF SEEN PERTURBED QUERIES

AdaR learns adaptive reasoning by comparing feedback on responses to perturbed queries whose problem-solving logic are consistent; consequently, the model should be exposed to these queries during the RLVR stage. We posit two situations where this comparison arises.

First, during SFT, memorization of superficial features associated with a given query induces spurious reasoning. When solving a perturbed query derived from that query in SFT, the model relies on spurious reasoning, producing unstable rollouts and thus yielding different feedback

Table 4: Effect of seen perturbed queries.

| Method | # Samples | In-Domain | Out-of-Domain |
|---|---|---|---|
| Standard-RLVR | 9K | 77.82 | 37.20 |
| AdaR-Lite×4 | 2.25K × 4 | 83.42 | 49.90 |
| AdaR | 9K | 84.72 | 53.88 |

for comparison. It's consistent with result shown in Table 4, training AdaR with perturbed data conditioned on SFT seed data leads to higher performance than "Standard-RLVR" which is trained on normal data.

Second, during RLVR, presenting the model with multiple perturbed queries that share consistent problem-solving logic but instantiate different valuable values can likewise induce comparison: if the model depends on spurious reasoning, it will receive different feedback across responses to these queries. To validate this, we additionally sample a 2.25K subset from the ORCA-MATH

dataset with no overlap with the SFT training data and apply 4 rounds of *controllable perturbation* for comparison. We denote this setting as "AdaR-Lite×4". As reported in Table 4, "AdaR-Lite×4" outperforms "Standard-RLVR", further corroborating this hypothesis.

### 4.5 AdaR is Applicable to Instruct Model

We further investigate using Qwen3-4B-Instruct-2507 (Yang et al., 2025) as the base model. AdaR still yields substantial performance gains (+5.86) and effectively mitigates robustness failures that persist under test-time scaling (+1.51). We additionally observe that the model's rollout time on seed dataset is shorter than that of AdaR ($\approx 0.36\times$), suggesting that generating correct answers becomes more difficult when the model is exposed to mild numerical perturbations, revealing pronounced brittleness. Taken together, these findings highlight a pervasive limitation in existing models: strong performance on conventional test sets often obscures fundamental weaknesses in generalization and robustness. In contrast, AdaR effectively exposes and remedies such latent vulnerabilities, demonstrating its broad applicability. Additional details are provided in Appendix A.6.

## 5 Related Work

**Analysis of Robustness and Generalization.** CoT has shown success in enhancing reasoning performance, with theoretical explanations suggesting CoT enables the chaining of accurate local inferences to estimate relationships between unseen variables (Prystawski et al., 2023; Feng et al., 2023). However, recent studies reveal significant robustness and generalization challenges in LLMs' mathematical capabilities. Wang et al. (2023) discovered that even invalid CoT demonstrations achieve comparable performance to valid ones. Similarly, we demonstrate that CoT arising from spurious reasoning is less sensitive to random variations in logical order. LLMs exhibit brittleness when facing problem variations: the GSM-SYM benchmark (Mirzadeh et al., 2024) demonstrates poor robustness on in-domain tasks with altered numerical values, while Jahin et al. (2025) shows limited generalization to out-of-domain problems, indicating models may memorize patterns rather than genuinely understand reasoning. The generation of GSM-SYM relies on human effort, and therefore only a small amount of data can be generated for testing. In contrast, AdaR is fully automatic, produces high-quality data, and can be used for both training and testing.

**Approaches to Improving Reasoning Ability.** Data synthesis has emerged as a promising solution (Wang et al., 2025b), with Li et al. (2023) paraphrasing questions to diversify templates, though showing limited improvement in adaptive reasoning due to unchanged mathematical structures. Lu et al. (2024) advances this by augmenting datasets through LLM-based generation and verification, helping models identify shared logical patterns through contrasting perturbed queries, but remains constrained by simple perturbations and error-prone verification. More importantly, we put forward a novel perspective, i.e. adaptive reasoning in subfigure I of Figure 1, which can explain why the aforementioned methods are effective. Although RLVR has been widely adopted to enhance model reasoning capabilities (Guo et al., 2025a; Team, 2025; Team et al., 2025) and improve generalization (Wen et al., 2025b; Wang et al., 2025a), recent research also reveals several critical limitations, such as format sensitivity (Huang et al., 2025), reward hack (Guo et al., 2025b), and degradation of reasoning capacity boundary (Yue et al., 2025a; Wu et al., 2025). Unlike existing optimization methods that focus primarily on algorithm-level improvements (Yu et al., 2025; Yue et al., 2025b; Zheng et al., 2025), AdaR adopts a complementary approach by inducing adaptive reasoning through the synthesis of high-quality perturbed data.

## 6 Conclusion

Robustness and generalization remain central challenges for LLMs when solving mathematical problems. We attribute these failures to spurious reasoning that relies on superficial features and encourage adaptive reasoning that can adapt to varying variable values. Therefore, we propose AdaR, a framework that enables adaptive reasoning and comprises a data synthesis component and a model training component. Experimental results demonstrate substantial performance improvements with a small amount of data on both in-domain and out-of-domain tasks. Further analyses indicate that AdaR indeed facilitates adaptive reasoning and is a scalable and broadly applicable framework.

## LIMITATIONS

Our framework is subject to several limitations. Firstly, although Section 4.3 underscores the importance of scaling L, AdaR currently enhances only T and x. We therefore identify scaling L-diversity as a promising direction for future work. Secondly, AdaR requires the entire problem-solving process of each seed queries to be expressible as a fully executable program, and it assumes the presence of easily perturbed input variables. These requirements implies applying AdaR in more general settings (e.g. theorem proving) would require more sophisticated designs. Thirdly, due to computational constraints, we do not investigate reasoning LLMs as base.

## REPRODUCIBILITY STATEMENT

We provide, in the supplementary materials, all settings necessary to reproduce our experimental results, including data synthesis scripts, training scripts and code, configuration files, and fixed random seeds. We will additionally release trained checkpoints to further facilitate reproduction.

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

# A APPENDIX

## A.1 THE USE OF LLMS

We use LLMs (e.g., GPT-5) only to polish writing. Specifically, their application focuses on two key areas: correcting grammatical errors and suggesting more appropriate word choices to enhance expression. Additionally, we conduct a thorough double check of all content refined by LLMs. This verification process is critical to preventing the inclusion of harmful information and ensuring the overall accuracy, reliability, and appropriateness of the final content.

## A.2 PROMPTS

We show the prompts used for generating the query template and problem-solving code in Prompt 1, Solution Existence in Prompt 2, code generation which evaluates reasoning ability in Prompt 3, code execution which evaluates computational ability in Prompt 4.

## A.3 GENERAL SETTINGS

### A.3.1 DATA SYNTHESIS

Given that most publicly available math word problem datasets are constructed based on GSM8K (Cobbe et al., 2021), and considering that existing LLMs may have been trained on substantial variations of this dataset, which alleviates to some extent the model's shortcut learning issues, we have opted for a newer and larger dataset ORCA-MATH (Mitra et al., 2024). We sample the data for synthesis by uniformly selecting using 42 as the random seed.

We use Qwen2.5-72B for data synthesis, including problem-solving codes, query templates, the existence of valid solutions, and implementations of other baselines. The temperature is set to 0.7, the top p is set to 0.95, and the maximum generation length is 4096 tokens.

## A.3.2 Training

All baselines are first trained with the same SFT procedure, after which RLVR training is performed using their respective data. For SFT, we use LLaMA-Factory (Zheng et al., 2024). All models are fine-tuned for 3 epochs with a batch size of 128 on 4 NVIDIA A100 GPUs. The peak learning rate is 1e-5 with a linear warm-up over the first 3% of training steps, followed by cosine decay to a minimum of 1e-7. The maximum generation length is set to 4096 tokens. For DAPO, we use veRL (Sheng et al., 2024), adopting the training setup of Yu et al. (2025). We utilize the AdamW optimizer (Loshchilov & Hutter, 2017) with a constant learning rate of 1e-6 and a linear warm-up over 20 rollout steps. For rollout, the prompt batch size is 256 and we sample 16 responses per prompt. For training, the mini-batch size is 32, corresponding to 128 gradient updates per rollout step.

## A.3.3 Evaluation

We compare AdaR with baselines on the following 7 benchmarks:

- **GSM8K** (Cobbe et al., 2021): The test set comprises 1,319 high-quality grade-school mathematics word problems, each requiring between 2 and 8 reasoning steps.

- **ORCA-AdaR-test**: The test set consists of 2,500 high-quality grade-school mathematics word problems synthesized by AdaR. Seed problems are selected from ORCA (Mitra et al., 2024) and subjected to 1-4 numerical perturbations to assess models' robustness in mathematical reasoning.

- **GSM-SYM** (Mirzadeh et al., 2024): The test set includes three subsets—*main*, *p1*, and *p2*—with 5,000, 5,000, and 2,500 instances, respectively. Starting from the GSM8K test problems, a symbolic template is constructed for each problem; each template yields 50 instances. The subsets augment the original logical structure by adding 0 (*main*), 1 (*p1*), or 2 (*p2*) additional reasoning clauses, thereby enabling a more rigorous evaluation of mathematical reasoning robustness.

- **MATH** (Hendrycks et al., 2021): The test set comprises 5,000 problems drawn from high-school mathematics competitions. Problems are categorized into seven types (Prealgebra, Intermediate Algebra, Algebra, Precalculus, Geometry, Counting & Probability, and Number Theory) and five difficulty levels.

- **CollegeMath** (Tang et al., 2024): The test set contains 2,818 college-level problems curated from nine college-level mathematics textbooks, covering seven key disciplines: Algebra, Precalculus, Calculus, VectorCalculus, Probability, LinearAlgebra, and Differential Equations.

- **TheoremQA** (Chen et al., 2023): A theorem-driven question-answering benchmark containing 800 problems grounded in 350 theorems, designed to evaluate LLMs' ability to apply domain-specific theorems across Mathematics, Physics, Electrical Engineering, Computer Science, and Finance.

- **AIME 2025**: A test set containing 30 problems from the 2025 American Invitational Mathematics Examination (AIME), curated to evaluate LLMs on challenging, Olympiad-level high-school mathematics across Algebra, Geometry, Number Theory, and Combinatorics.

We adopt the evaluation pipeline of Tong et al. (2024) with some modifications. Unless otherwise noted, we set the sampling temperature to 0.7 and the nucleus parameter top_p to 0.9. Regarding evaluation metrics, pass@1 is defined as the accuracy of the first sampled output, whereas avg@32 is the mean accuracy computed over 32 sampled outputs.

## A.4 Structural Code Snippet

To calculate the frequency of structural code snippet in the responses, we inspect 20 responses to queries from GSM-SYM. We present cases of structural code snippets from "Initial-SFT" and AdaR in Figure 8.

## A.5 GENERATION COST

The generation costs for all aspects included in the *sanity check* are reported in Table 5. This experiment is conducted on a single compute node.

## A.6 PERFORMANCE ON INSTRUCT MODELS

**Dataset.** To evaluate whether AdaR remains effective on more challenging datasets, beyond the grade-school mathematics problems used in earlier experiments (i.e., ORCA dataset), we adopt the Light-R1 dataset (Wen et al., 2025a). From this dataset, we sample 7.5K seed data and apply numerical perturbations 1–4 times, resulting in a 17K-instance training set, denoted as "R1-AdaR-train". To perform a fine-grained analysis of whether AdaR improves model robustness and to avoid potential of data leakage, we construct two test sets, each containing 1K instances: "seen", whose query templates appear in "R1-AdaR-train", and "unseen", whose templates do not appear in the "R1-AdaR-train".

**Baselines.** In this experiment, we introduce two strong baselines: "Qwen3-4B-Instruct-2507" and the test-time-scaling-enabled "Qwen3-4B-Thinking-2507". Because the Instruct model already possesses strong mathematical reasoning capabilities, we skip the SFT stage and directly perform RLVR training. The model trained using RLVR on the 7.5K seed data is referred to as "Standard-RLVR-Instruct", while the model trained with AdaR synthetic data is denoted "AdaR-Instruct".

Table 6 shows the full results.

## A.7 TRAINING DYNAMIC ANALYSIS

As shown in Figure 5 and Figure 6, during RLVR training, as the training steps increase, the model's generated CoT sequences grow progressively longer. In the early stage, although the sequence length steadily increases, the model rapidly acquires stable short reasoning patterns, often accompanied by certain forms of spurious reasoning, resulting in a decrease in entropy. As training continues and CoT sequences further lengthen, the model begins to develop more adaptive reasoning due to exposure to perturbed data. This transition increases the model's predictive uncertainty, leading to the subsequent rise in entropy.

In parallel, Figure 7 shows that the AIME 2024 scores exhibits a rapid improvement during the initial training stage and then plateaus with minor fluctuations in later steps. This trajectory aligns with entropy dynamics: the model first stabilizes around short reasoning patterns that yield quick performance gains, and later explores a broader reasoning space as CoT sequences grow, leading to small oscillations around a higher performance level.

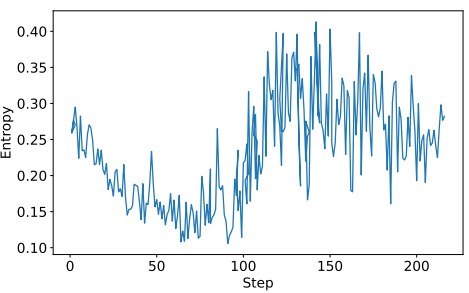

Figure 5: Entropy of actor model.

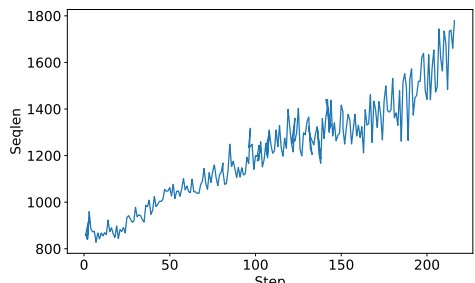

Figure 6: Generated sequence length of actor model.

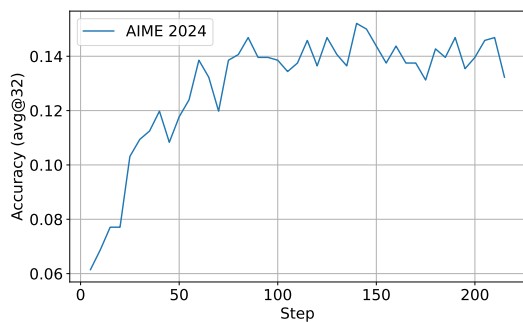

Figure 7: AIME 2024 scores on AdaR during training.

Table 5: Generation cost of component in *sanity check*.

| Sanity Check | Generation Times (s per 1K samples) | Retention rate (%) |
|---|---|---|
| Variable Alignment | 2.90 | 67.70 |
| Executable Code | 2.80 | 70.70 |
| Solution Existence | 612 | 95.17 |

---

**Prompt 1: Generate Template and Code**

Task Description:
You are given a natural language query and its chain-of-thought response.
Your task is to: Generate a Query Template by abstracting specific values into variables.
Generate Python Code that executes the logic described in the COT response using the abstracted variables.

Input Format:
Query: Original query with specific values
Response: Chain-of-thought reasoning that leads to the answer

Output Requirements:
Query Template:
Replace only concrete values in the query with angle-bracketed placeholders like <variable_name>. Do not replace names or general nouns (e.g., do not change "Jungkook" to <person_name>). Preserve the original wording and structure of the query as much as possible.
Python Code:
Begin by defining variables that correspond to the placeholders in the template. Translate the logic in the response into executable Python code. The code should end with a print() statement that prints only the final result. Do not include comments with explanations or reasoning. Use the same variable names as in the template for consistency.

=== START EXAMPLE ===

{example}

=== END EXAMPLE ===

### Query:
{query}

### Response:
{response}

$q_i$: A group of 157 students has various hobbies. 23 like to play video games, 41 like to play volleyball, and the rest like to either hike or paint. How many like to hike if the number that like to paint is twice the number that prefer playing volleyball?

$y_i$: 11

| $z_i$ from "Inital-SFT": | $z_i$ from AdaR: |
|---|---|
| ... we multiply the number of volleyball players by 2:

Number of volleyball players = 41
Number of students who like to paint = 2 * 41 = 82

...
We subtract the sum of these from the total number of students to find out how many like to hike

...
Number of students who like to hike = 157 - (23 + 41 + 82)

...
Number of students who like to hike = 11... | Let's denote the number of students who like to bake as \( B \) and the number of students who like to hike as \( H \)
...
The number of students who like to hike is twice the number that prefer playing football, so \( H = 2 \times 41 = 82 \)
...
Therefore, the number of students who like to bake and hike combined is the total number of students minus the number of students who like to paint and play football.

So, we can write the equation:
\[ B + H = 157 - (23 + 41) \]

Substituting the value of \( H \) into the equation, we get:\n\[ B + 82 = 157 - (23 + 41) \]
\[ B + 82 = 157 - 64 \]
\[ B + 82 = 93 \]
Now, we solve for \( B \):
\[ B = 83 - 82 \]\n\[ B = 11 \]... |

$q_j$: A group of 183 students has various hobbies. 42 like to play video games, 25 like to play volleyball, and the rest like to either hike or paint. How many like to hike if the number that like to paint is twice the number that prefer playing volleyball?

$y_j$: 66

| $z_j$ from "Inital-SFT": | $z_j$ from AdaR: |
|---|---|
| ... Since the number of students who like to paint is twice the number that prefer playing tennis, we can calculate

Number of students who like to paint = 2 * 25 = 50
...
We can subtract these from the total to find out how many like to play music

...
Number of students who like to play music = 183 - (42 + 50)

...
Number of students who like to play music = 91... | Let's denote the number of students who like to play music as \( M \) and the number of students who like to paint as \( P \)
...
The number of students who like to paint is twice the number that prefer playing tennis, so \( P = 2 \times 25 = 50 \)
...
Therefore, the number of students who like to play music or paint is the total number of students minus the number of students who like to bake and the number of students who like to play tennis. This can be expressed as:
\[ M + P = 183 - 42 - 25 \]

Substituting the value of \( P \) into the equation, we get:
\[ M + 50 = 183 - 42 - 25 \]

Now, let's calculate the right-hand side of the equation:
\[ 183 - 42 = 141 \]
\[ 141 - 25 = 116 \]

So, we have:
\[ M + 50 = 116 \]

To find \( M \), we subtract 50 from both sides of the equation:
\[ M = 116 - 50 \]
\[ M = 66 \]... |

Figure 8: The case study of structural text in outputs. The green background indicates the correct reasoning step. The red background indicates the wrong reasoning step.

Table 6: Comparison with the Instruct model and Thinking model.

| Method | Robustness | | | Generalization | | | | | AVG | CoT Length |
|---|---|---|---|---|---|---|---|---|---|---|
| | seen | unseen | GSM-SYM | GSM8K | MATH | College | Theorem | AIME | | |
| Qwen3-4B-Instruct-2507 | 60.99 | 56.62 | 85.50 | 92.27 | 91.12 | 50.85 | 47.50 | 46.46 | 66.41 | 1672 |
| Qwen3-4B-Thinking-2507 | 66.32 | 63.42 | 91.19 | 93.33 | **95.38** | **53.97** | **58.13** | **75.83** | **74.70** | 4784 |
| Standard-RLVR-Instruct | 62.86 | 57.72 | 90.47 | 93.78 | 91.52 | 52.09 | 48.25 | 52.50 | 68.65 | 2355 |
| AdaR-Instruct | **68.42** | **65.17** | **91.88** | **94.69** | 90.83 | 53.30 | 55.00 | 58.83 | 72.27 | 3435 |

---

### Prompt 2: Existence of Valid Solution

Task Description:
Your task is to generate a Chain-of-Thought (CoT) explanation that answers the user's question by reasoning through the logic implied in a provided Python script. Use the script to inform your explanation, but do not output or reproduce any code.

Input Format:
Query: A question involving specific values or conditions.
Python Code: A script that solves the query or provides a key computational procedure.

Output Requirements:
Start by interpreting the question clearly. Reason through the problem step by step, using the Python code as a guide to inform your logic. Refer to relevant steps in the code as part of your reasoning. Do not output or reference the code in any form. Explicitly state the final answer after the final step within \boxed{}.

### Query:
{query}

### Python Code:
{code}

---

### Prompt 3: Code Generation

Please write a Python code to solve the following problem. Just give me the code, no explanation, no comments, no input statements. The code should be runnable and print the answer in the end.

### Query:
{query}

### Python Code:

---

### Prompt 4: Code Execution

Please help me run the following Python code and return its output result instead of the code itself:
{code}

