# OpenReview forum: "Make Mathematical Reasoning Adaptive"
_ICLR.cc/2026/Conference — ICLR 2026 Conference Withdrawn Submission_

### Official Review · Reviewer_k8Au · 2025-10-21

**Soundness:** 2
**Presentation:** 3
**Contribution:** 2
**Rating:** 4
**Confidence:** 4

**Summary:**

This paper proposes a method for synthesizing mathematical reasoning training data to expand verifiable training samples for reinforcement learning. The method transforms existing mathematical reasoning data into a problem template and corresponding answer-generating code. By introducing controllable perturbations, it generates a set of similar problems with varying conditions. The model trained with reinforcement learning on the constructed dataset achieves superior performance on multiple benchmarks compared with direct SFT, RLVR, and other synthetic data approaches, demonstrating the effectiveness of the proposed method.

**Strengths:**

**[S1]** The idea of converting problems into templates and executable code is reasonable, as it allows for controlled generation of diverse training data.
**[S2]** The paper is clearly written, and the proposed method is easy to follow.
**[S3]** The paper provides detailed descriptions of the experimental setup and releases code, demonstrating good reproducibility.

**Weaknesses:**

**[W1] Unsubstantiated claim:** In Section 2.2, the paper claims that SFT is not suitable for training. This statement appears anecdotal and lacks empirical evidence. The authors should provide experiments to substantiate this claim.

**[W2] Unclear data sampling strategy:** While ORCA-MATH contains 200k samples, this paper only uses 9k of them. However, the authors did not explain how these 9k samples were selected. A detailed description of the sampling strategy is necessary.

**[W3] Inappropriate title:** The current title “Making Mathematical Reasoning Adaptive” does not accurately reflect the adaptive nature of the proposed method. The authors should reconsider their title to better align with the paper’s actual contributions.

**[W4] Inconsistent performance gains:** Compared with Standard-RLVR, AdaR achieves about a 10% average improvement on Qwen2.5-MATH, but only ~2% on DeepSeekMath and ~3% on Llama3. The authors should provide a detailed explanation for these discrepancies, especially regarding how the choice of base model affects performance gains.

**[W5] Evaluation stability:** I appreciate that the authors evaluated avg@32 on AIME25 to reduce evaluation variance. However, similar evaluations should also be conducted on other datasets to ensure stability and provide more reliable conclusions.

**[W6] Lack of training dynamics analysis:** In reinforcement learning, training dynamics are often more informative than a single performance number. The paper does not analyze training dynamics, which limits the insights provided to the reader.

**[W7] Lack of comparison with existing work:** The idea of constructing problems from templates has been explored in prior work [1]. The authors should clarify the key differences between their approach and existing methods.


Ref:
[1] Mirzadeh et al. GSM-Symbolic: Understanding the Limitations of Mathematical Reasoning in Large Language Models. ICLR 2025.

**Questions:**

**[Q1]** In Table 2, the fourth row is empty. Did the authors forget to include the experimental results?

**[Q2]** For the baseline experiments, were MetaMath and MathGenie trained using SFT or RLVR?

**[Q3]** AdaR generates more than 9k data samples through templates. In Table 1, how was data volume controlled for fair comparison? Did AdaR and Standard-RLVR use the same number of training samples and RL training steps?

**[Q4]** Could the authors provide some examples of templates and codes? This would help readers better understand the practical effect of the proposed method.

---

> ### Author Response · Authors · 2025-11-27
>
> # [W1] Unsubstantiated claim
> We had cited [1] and [2] in Section 2.2 to support the claim that SFT tends to memorize reasoning CoTs and fall into a local optimum. Our intention is not to deny the usefulness of SFT during training at all. Rather, we argue that such memorization is inherently limited and can exhibit deficiencies, which motivates our further analysis.
>
> # [W2] Unclear data sampling strategy
> We acknowledge that the sampling details are not explicitly stated in the original submission. In fact, we uniformly sample from the Orca dataset using seed=42. This clarification has been added to the updated PDF.
>
> # [W3] Inappropriate title
> Please refer to the section "Importance of the Adaptive Reasoning Formulation" in the *Overall Response*.
>
> # [W4] Inconsistent performance gains
> As discussed in Observation 3 of Section 3.2, we hypothesized that the model must possess sufficiently strong mathematical pre-training in order to generate correct rollouts during RL. To support this claim, we further report—now included in the updated PDF—the number of additional rollout problems required for each model to obtain both correct and incorrect answers at batch size 256: Qwen (458.00), DeepSeekMath (563.20), and LLaMA3 (942.83), which are inversely correlated with their respective performance gains.
>
> # [W5] Evaluation stability
> As stated in Appendix 3.3, our evaluation setup follows [2], with minor adaptations to accommodate different chat templates. We use avg@32 for AIME because the dataset contains only 30 samples. In contrast, the other datasets contain roughly 1,000 samples, for which variance concerns are considerably smaller.
>
> # [W6] Lack of training dynamic analysis
> We have added a Training Dynamic Analysis section in Appendix 6. During RLVR training, as the training steps increase, the model’s generated CoT sequences grow progressively longer. In the early stage, although the sequence length steadily increases, the model rapidly acquires stable short-range reasoning patterns—often accompanied by certain forms of spurious reasoning, resulting in a decrease in entropy. As training continues and CoT sequences further lengthen, the model begins to develop more adaptive reasoning due to exposure to perturbed data. This transition increases the model’s predictive uncertainty, leading to the subsequent rise in entropy.
>
> # [W7] Lack of comparison with existing work
> Please refer to the section "Innovation of the AdaR Training Framework" in the *Overall Response*.
>
> # [Q1] In Table 2, the fourth row is empty. Did the authors forget to include the experimental results?
> As explained in the second paragraph of Section 3.3, ablating the EC (Executable Code) filter results in too few remaining samples to support a complete experiment.
>
> # [Q2] For the baseline experiments, were MetaMath and MathGenie trained using SFT or RLVR?
> We apologize for not making this explicit in the experimental setup. To ensure consistency across conditions, all methods are trained using the RLVR training pipeline.
>
> # [Q3] AdaR generates more than 9k data samples through templates. In Table 1, how was data volume controlled for fair comparison? Did AdaR and Standard-RLVR use the same number of training samples and RL training steps?
> For the first question, as described in the Data Synthesis subsection of Section 3.1, for each data in seed data, we select one corresponding synthetic data to construct the ORCA-AdaR-train.
>
> For the second question, as stated in Data Synthesis and Baseline subsection (in Section 3.1), both AdaR and Standard-RLVR are trained on the same 9K synthesized dataset. Appendix 3.2 reported the epoch and batch size settings for both configurations. Section 3.1 also stated that the dataset sizes are identical across the two settings; therefore, the total number of training steps remains the same.
> [Q4] Examples of templates and codes for clarity
> Subfigure (I) in Figure 1 had presented straightforward illustrative examples of both the query template and the problem-solving logic as code. For the problem
>
> $q$: Find A such that 32 × A × A × A = 42592,
>
> the corresponding query template is
>
> $T$: Find A that satisfies \<coefficient\> × A × A × A = \<result\>,
>
> and the associated problem-solving logic as code is:
> ```python
> A_cubed = result / coefficient
> A = A cubed ** (1 / 3)
> ```
>
>
> [1] Sft memorizes, rl generalizes: A comparative study of foundation model post-training.
>
> [2] Quagmires in SFT-RL Post-Training: When High SFT Scores Mislead and What to Use Instead
>
> [3] Dart-math: Difficulty-aware rejection tuning for mathematical problem-solving
>
> [4] Gsm-symbolic: Understanding the limitations of mathematical reasoning in large language models.
>
> [5] Metamath: Bootstrap your own mathematical questions for large language models.
>
> [6] Mathgenie: Generating synthetic data with question back-translation for enhancing mathematical reasoning of llms.

---

> > ### Comment · Reviewer_k8Au · 2025-11-27
> >
> > I appreciate the authors’ comprehensive rebuttal and the revisions made to the manuscript. However, several points still remain unclear to me:
> >
> > > W2:  In fact, we uniformly sample from the Orca dataset using seed=42.
> >
> > Could the authors clarify the motivation behind this down-sampling strategy? Why not directly utilize the full 200K OrcaMath dataset?
> >
> > > W3: Please refer to the section "Importance of the Adaptive Reasoning Formulation" in the Overall Response.
> >
> > After carefully reading the explanation, I still find the paper’s title potentially inappropriate. The term “Adaptive Reasoning Formulation” suggests a specific methodological training framework, whereas the title “Make Reasoning Adaptive” implies a broader or different focus. Could the authors better justify why these two notions should be viewed as consistent? Or changing your title to better reflect the contribution of you work.
> >
> > > W5:  the other datasets contain roughly 1,000 samples, for which variance concerns are considerably smaller.
> >
> > While it is reasonable to follow existing evaluation protocols, many prior works also report avg@32 for these benchmarks. Could you provide data-based evidence to demonstrate that there is no inconsistency between avg@4 and avg@32?
> >
> > > W6: We have added a Training Dynamic Analysis section in Appendix 6. During RLVR training.
> >
> > Would it be possible for the authors to include accuracy curves over training steps on representative benchmarks (e.g., AIME or MATH)? This would help readers better understand the model’s learning dynamics and performance progression.

---

> ### Author Response · Authors · 2025-11-29
>
> We sincerely thank the reviewer for the continued careful reading of our rebuttal and for raising these additional questions.  Below, we address each point in detail.
>
> > W2: Could the authors clarify the motivation behind this down-sampling strategy?
>
> Given that utilizing the full 200K dataset would incur prohibitive computational costs, we opted for a reduced 9K training set. Concretely, RLVR training on the current 9K subset already requires approximately **2 days** to complete on 8×A100 GPUs. Scaling to the full 200K examples, roughly 22× larger, would extend the duration of a single baseline experiment to more than **44 days**, which is infeasible under our computational constraints.
>
> The dataset size, 9K,  is motivated by two considerations. First, this scale results in a training time of roughly 2–3 days, which is acceptable for conducting multiple experimental runs. Second, the dataset size is comparable to that used in DAPO [1], which is likewise a RL related work. Moreover, as discussed in Section 4.2, we empirically evaluated the marginal gains obtained by increasing the amount of data—that is, increasing $L$-diversity. Therefore, we believe that the 9K dataset is sufficient to validate the effectiveness of our method.
>
> > W3: After carefully reading the explanation, I still find the paper’s title potentially inappropriate.
>
> After careful consideration, we agree that the original title, "Make Mathematical Reasoning Adaptive," could inadvertently suggest that our primary contribution lies in adaptive thinking (i.e., approaches that enable models to adjust chain-of-thought lengths according to problem difficulty). To avoid this potential misinterpretation, we propose the revised title:
>
> **"AdaR: A Framework for Equipping LLMs with Adaptive Reasoning"**,
>
> which more accurately reflects our contribution.
>
> > W5: While it is reasonable to follow existing evaluation protocols, many prior works also report avg@32 for these benchmarks.
>
> Due to constraints in time and computational resources, we report **"avg@1 / avg@32"** results in each cell for a subset of models evaluated on a subset of test set used in our paper. As shown, the performance differences remain within 1 percentage point and are **fully consistent with our conclusions drawn from the avg@1 results**.
>
> | Method            | GSM8K           |    GSM-SYM-main        | MATH        |
> |-----------------|-----------------|-----------------|-----------------|
> | Initial SFT     | 81.43 / 80.42   | 74.26 / 73.89   | 42.84 / 42.57   |
> | Standard-RLVR   | 86.96 / 86.88   | 81.34 / 81.51   | 61.71 / 61.29   |
> | MetaMATH        | 84.61 / 83.80   | 79.78 / 78.88   | 66.92 / 66.29   |
> | AdaR            | 91.81 / 92.32   | 89.72 / 89.39   | 75.90 / 76.00   |
>
> > W6: Would it be possible for the authors to include accuracy curves over training steps on representative benchmarks (e.g., AIME or MATH)?
>
> We thank you for your suggestion and have added the performance trajectory of the model on the AIME 2024 test set throughout the training process in Appendix 6.
>
> Figure 7 shows that the AIME 2024 scores exhibit **a rapid improvement during the initial training stage and then plateau with minor fluctuations in later steps**. This trajectory aligns with entropy dynamics: the model first stabilizes around short reasoning patterns that yield quick performance gains, and later explores a broader reasoning space as CoT sequences grow, leading to small oscillations around a higher performance level.
>
> [1] DAPO: An Open-Source LLM Reinforcement Learning System at Scale

---

### Official Review · Reviewer_prZi · 2025-10-29

**Soundness:** 2
**Presentation:** 3
**Contribution:** 2
**Rating:** 4
**Confidence:** 4

**Summary:**

The paper proposes the AdaR framework to enhance the mathematical reasoning abilities of LLMs. The core problem AdaR addresses is the models' tendency toward spurious reasoning, where they rely on superficial features rather than the actual problem-solving logic, leading to failures in robustness and generalization when variable values change.

The AdaR framework works by enabling adaptive reasoning, which allows LLMs to adapt to varying variable values when the underlying problem-solving logic is preserved. This is achieved through two main contributions: First, a data synthesis method generates logically equivalent problems by controllably perturbing the numerical values in the variable set. The ground-truth answers are obtained by code execution and validated to ensure high-quality data. Second, a training strategy uses RLVR on this synthetic data. This approach penalizes models that exhibit spurious reasoning by failing on the perturbed queries, thereby compelling them to learn the genuine, adaptive problem-solving logic.

Analysis confirms that AdaR successfully induces adaptive reasoning by improving models' algebraic thinking and increasing their adherence to logical order.

**Strengths:**

* The paper introduces a new framework which directly targets and mitigates the problem of spurious reasoning in LLMs. This focused approach enhances the robustness and generalization of LLMs in mathematical problem-solving tasks.

* A new data synthesis method that generates logically equivalent problems by perturbing numerical values, combined with sanity check. This ensures the training data accurately reflects a preserved problem-solving logic.

**Weaknesses:**

* The data synthesis approach, which involves perturbing numerical values to create logically equivalent problems, is not fundamentally novel. Similar methods, such as those discussed in the GSM-Symbolic paper [1], have previously highlighted models' lack of robustness to simple numerical variations. Although the authors claim their method eliminates human annotation, this is not a substantial innovation.

* The evaluation is not fully convincing because the performance gains are demonstrated on non-reasoning models, and scores on hard benchmarks like AIME remain quite low. It is highly probable that the benefits of AdaR could diminish when compared against test-time scaling or models trained on significantly larger datasets.

References:

[1] GSM-Symbolic: Understanding the Limitations of Mathematical Reasoning in Large Language Models

**Questions:**

1. The paper title is kind of confusing, the goal is to make LLMs more robust and generalizable in mathematical reasoning. I do not see how "adaptive reasoning" fits in here.

---

> ### Author Response · Authors · 2025-11-27
>
> # [W1] Limited Innovation
> Please refer to the section "Innovation of the AdaR Training Framework" in the *Overall Response*.
>
> # [W2] Lack of strong baseline
> To substantiate the presence of robustness issues in current reasoning models, we conducted additional diagnostic experiments. To avoid any risk of data leakage, we evaluated both Qwen3-4B-Instruct-2507 and Qwen3-4B-Thinking-2507 [1] on our constructed test set , ORCA-test-unseen (consisting of primary-school math problems). It is important to clarify that, unlike the AdaR-test set used in the main paper, the query templates in ORCA-test-unseen never appear in the AdaR training set.
>
> We observe that Qwen3-4B-Thinking-2507, even after post-training on massive data and further relying on test-time scaling, still fails to fully resolve robustness issues. Its performance is even slightly weaker than our model trained on only 10K examples and built on weaker base model Qwen2.5-Math-7B.
>
> | Model | ORCA-test-unseen |
> |-|:-:|
> | Qwen3-4B-Instruct-2507 | 80.92 |
> | Qwen3-4B-Thinking-2507 | 82.62 |
> | AdaR | 83.40 |
>
> To further examine whether AdaR can compensate for the robustness issues of these stronger models, we have additionally conducted experiments using Qwen3-4B-Instruct-2507 on dataset LightR1[2]. However, because this model typically requires approximately 10K-token CoT on our synthetic data, running this model under the same experimental setting as before is computationally expensive (approximately 2-3 hours per training step). As a result, the corresponding results are not yet available; **training is expected to finish in approximately 2 days, and we will upload the results immediately once they are produced.**
>
> We do not include experiments on reasoning models because it generally requires even longer CoTs, , which would require more than **18 days** to complete approximately 150 training steps, making it infeasible to complete the training within the rebuttal period.
>
> # [Q1] Confusion on the relationship between "adaptive reasoning"  and our goal
> We have addressed this question in the *Overall Response*.
>
> [1] Qwen3 technical report
>
> [2] Light-r1: Curriculum sft, dpo and rl for long cot from scratch and beyond.

---

> > ### Author Response · Authors · 2025-12-01
> > **Experiment for W2**
> >
> > # AdaR effectively mitigates robustness deficiencies that persist under test-time scaling.
> >
> > The results are presented in the table below. "Standard-RLVR-Instruct" denotes training directly on the seed data, whereas "AdaR-Instruct" refers to training on synthetic data generated by AdaR.  Both settings apply RLVR training to "Qwen3-4B-Instruct-2507". The **"seen"**/**"unseen"** correspond to test sets whose query templates in Light-R1 are observed / unobserved during training, respectively.
> >
> > | Method                 | seen      | unseen    | GSM-SYM   | GSM8K     | MATH      | College   | Theorem   | AIME      | AVG       |
> > | ---------------------- | --------- | --------- | --------- | --------- | --------- | --------- | --------- | --------- | --------- |
> > | Qwen3-4B-Instruct-2507 | 60.99     | 56.62     | 85.50     | 92.27     | 91.12     | 50.85     | 47.50     | 46.46     | 66.41     |
> > | Qwen3-4B-Thinking-2507 | 66.32     | 63.42     | 91.19     | 93.33     | **95.38** | **53.97** | **58.13** | **75.83** | **74.70** |
> > | Standard-RLVR-Instruct | 62.86     | 57.72     | 90.47     | 93.78     | 91.52     | 52.09     | 48.25     | 52.50     | 68.65     |
> > | AdaR-Instruct          | **68.42** | **65.17** | **91.88** | **94.69** | 90.83     | 53.30     | 55.00     | 58.83     | 72.27     |
> >
> > The result shows that **AdaR still yields substantial performance gains (+5.86) and effectively mitigates robustness failures that persist under test-time-scaling-enabled "Qwen3-4B-Thinking-2507" (+1.51)**.
> > We additionally observe that the model's rollout time on seed dataset is shorter than that of AdaR ($\approx0.36\times$), suggesting that generating correct answers becomes more difficult when the model is exposed to mild numerical perturbations, revealing pronounced brittleness.
> > Taken together, these findings highlight a pervasive limitation in existing models: **strong performance on conventional test sets often obscures fundamental weaknesses in generalization and robustness.**
> > In contrast, AdaR effectively exposes and remedies such latent vulnerabilities, demonstrating its broad applicability.
> > Additional details are provided in Appendix 6 in the updated PDF.
> >
> > *Owing to the inclusion of additional baselines and an unexpected computing resource failure, we regret that the experimental results on the strong baseline are delivered 2 days later than originally anticipated.*

---

### Official Review · Reviewer_Rq4c · 2025-10-30

**Soundness:** 2
**Presentation:** 2
**Contribution:** 2
**Rating:** 4
**Confidence:** 4

**Summary:**

This paper investigates the robustness and generalization failure of large language models (LLMs) in mathematical reasoning, attributing these issues to *spurious reasoning* — the tendency of models to rely on superficial features rather than true problem-solving logic.
To address this, the authors propose **AdaR**, a framework designed to induce *adaptive reasoning* through two key components:
1. **Data Synthesis** – generating logic-equivalent problems by perturbing variable values while keeping the underlying reasoning structure fixed.
2. **Reinforcement Learning with Verifiable Rewards (RLVR)** – using correctness feedback on perturbed problems to penalize spurious reasoning and reward adaptive logic.

Experiments on several mathematical reasoning benchmarks (e.g., GSM8K, ORCA-MATH, GSM-SYM, MATH, AIME) show consistent performance improvements (+8.5 points on average) across different base models (Qwen2.5-Math, DeepSeekMath, LLaMA3).

**Strengths:**

1. **Clear motivation** – The paper addresses an important and well-recognized problem: the lack of robustness and generalization in LLMs’ mathematical reasoning.
2. **Intuitive framework** – The idea of perturbing variable values while preserving logic is intuitive and well-illustrated.
3. **Automatic pipeline** – The data synthesis process is largely automated and uses executable code for answer verification, reducing annotation cost.
4. **Interesting metric proposal** – The introduction of “Influence to Logical Order (ILO)” offers a creative, though not fully formal, metric for reasoning robustness.

**Weaknesses:**

1. **Limited novelty** – The main idea of combining variable perturbation with RLVR is incremental compared to prior works such as *MetaMath* and *MathGenie*. The notion of “adaptive reasoning” lacks formal definition and mainly reframes existing robustness concepts.

2. **Insufficient theoretical and empirical grounding** – The paper does not provide a rigorous formulation or proof that AdaR truly induces adaptive reasoning. Experiments focus only on numeric perturbations, without testing structural or logical variations.

3. **Lack of strong baselines (Major)** – The paper only compares with relatively early baselines such as *MetaMath* and *MathGenie*. However, after the release of *DeepSeek-R1* and other strong RLVR-based reasoning frameworks in 2025, several more competitive datasets and methods have become available. A fair comparison with these recent baselines is essential to convincingly demonstrate the contribution and significance of the proposed dataset.

4. **Scalability and practicality concerns** – The Sanity Check process, especially the validation of solution existence, is computationally expensive (~600s per 1K samples), raising doubts about the framework’s scalability to larger datasets.

**Questions:**

1. How does the AdaR dataset compare with datasets like Synthetic-1 or BigMath?

2. How much of the observed improvement arises from the data synthesis itself versus the RLVR training process? Would SFT on the same synthetic data yield similar gains?

---

> ### Author Response · Authors · 2025-11-27
>
> # [W1] Limited novelty
> Please refer to the section "Innovation of the AdaR Training Framework" in the *Overall Response*, where we explicitly articulate the distinctions between our approach and prior works.
>
> Please refer to the section "Importance of the Adaptive Reasoning Formulation" in the *Overall Response* provides a more detailed explanation of the conceptual motivation and benefits underlying our Adaptive Reasoning formulation.
>
> # [W2] Insufficient theoretical and empirical grounding
> In Section 4.1, we present two complementary **empirical** perspectives intended to demonstrate that the model acquires adaptive reasoning. Additionally, in the ablation study (Section 3.3) and the scaling analysis (Section 4.3), we not only focused in numeric perturbation, but we explicitly evaluate how perturbation in query templates and problem-solving logics affects model's performance.
>
> A concrete theoretical grounding, however, lies beyond the scope of this work.
>
> # [W3] Lack of strong baseline
> To substantiate the presence of robustness issues in current reasoning models, we conducted additional diagnostic experiments. To avoid any risk of data leakage, we evaluated both Qwen3-4B-Instruct-2507 and Qwen3-4B-Thinking-2507 [3] on our constructed test set , ORCA-test-unseen (consisting of primary-school math problems). It is important to clarify that, unlike the ORCA-AdaR-test set used in the main paper, the query templates in ORCA-test-unseen never appear in the AdaR training set.
>
> We observe that Qwen3-4B-Thinking-2507, even after post-training on massive data and further relying on test-time scaling, still fails to fully resolve robustness issues. Its performance is even slightly weaker than our model trained on only 10K examples and built on weaker base model Qwen2.5-Math-7B.
>
> | Model | ORCA-test-unseen |
> |-|:-:|
> | Qwen3-4B-Instruct-2507 | 80.92 |
> | Qwen3-4B-Thinking-2507 | 82.62 |
> | AdaR | 83.40 |
>
> To further examine whether AdaR can compensate for the robustness issues of these stronger models, we have additionally conducted experiments using Qwen3-4B-Instruct-2507 on dataset LightR1[4]. However, because this model typically requires approximately 10K-token CoT on our synthetic data, running this model under the same experimental setting as before is computationally expensive (approximately 2-3 hours per training step). As a result, the corresponding results are not yet available; **training is expected to finish in approximately 2 days, and we will upload the results immediately once they are produced.**
>
> We do not include experiments on reasoning models because it generally requires even longer CoTs, , which would require more than **18 days** to complete approximately 150 training steps, making it infeasible to complete the training within the rebuttal period.
>
> # [W4] Scalability and practicality concerns
> We apologize for the misunderstanding regarding the actual cost of the sanity check. In fact, the reported overhead corresponds to local single-machine testing (we have updated this detail in the revised PDF). In practical deployments, efficiency can be substantially improved by using sglang to distribute the server across multiple machines. Based on our measurements on a 4-machine deployment, the generation cost decreases from 612 s per 1K samples to 172 s per 1K samples.
>
> # [Q1] Including experiments on additional datasets
> Synthetic-1 [5] rely on distilling correct CoTs from LLMs such as R1 to construct SFT datasets. And Big-Math [6] rely on human efforts. Compared with our method, these dataset primarily increase the diversity of $L$ (as discussed in Section 4.3). As we noted in the *Overall Response*, AdaR additionally complements this by increasing the diversity of $x$ and $T$, and therefore can be directly applied to models already trained on Synthetic-1 or BigMath synthetic data. This integration further is likely to enhance the model’s reasoning robustness and generalization capability.
>
> # [Q2] How much of the observed improvement arises from data synthesis? Would SFT on the same synthetic data yield similar gains?
> The improvement indeed arises from the data synthesis itself. Section 3.2 demonstrates such improvements (+4.39) by comparing the Standard-RLVR setting (where RLVR training data is derived from the Orca dataset, not used by SFT training). Moreover,  gains from SFT (+3.76) on the same synthetic data can be calculated by the ablation results in Section 3.3 under the AdaR-RFT setting.

---

> > ### Author Response · Authors · 2025-11-27
> >
> > [1] Metamath: Bootstrap your own mathematical questions for large language models.
> >
> > [2] Mathgenie: Generating synthetic data with question back-translation for enhancing mathematical reasoning of llms.
> >
> > [3] Qwen3 technical report
> >
> > [4] Light-r1: Curriculum sft, dpo and rl for long cot from scratch and beyond.
> >
> > [5] SYNTHETIC-1: Two Million Collaboratively Generated Reasoning Traces from Deepseek-R1
> >
> > [6] Big-Math: A Large-Scale, High-Quality Math Dataset for Reinforcement Learning in Language Models

---

> ### Author Response · Authors · 2025-12-01
> **Experiment for W3**
>
> # AdaR effectively mitigates robustness deficiencies that persist under test-time scaling.
>
> The results are presented in the table below. "Standard-RLVR-Instruct" denotes training directly on the seed data, whereas "AdaR-Instruct" refers to training on synthetic data generated by AdaR.  Both settings apply RLVR training to "Qwen3-4B-Instruct-2507". The **"seen"**/**"unseen"** correspond to test sets whose query templates in Light-R1 are observed / unobserved during training, respectively.
>
> | Method                 | seen      | unseen    | GSM-SYM   | GSM8K     | MATH      | College   | Theorem   | AIME      | AVG       |
> | ---------------------- | --------- | --------- | --------- | --------- | --------- | --------- | --------- | --------- | --------- |
> | Qwen3-4B-Instruct-2507 | 60.99     | 56.62     | 85.50     | 92.27     | 91.12     | 50.85     | 47.50     | 46.46     | 66.41     |
> | Qwen3-4B-Thinking-2507 | 66.32     | 63.42     | 91.19     | 93.33     | **95.38** | **53.97** | **58.13** | **75.83** | **74.70** |
> | Standard-RLVR-Instruct | 62.86     | 57.72     | 90.47     | 93.78     | 91.52     | 52.09     | 48.25     | 52.50     | 68.65     |
> | AdaR-Instruct          | **68.42** | **65.17** | **91.88** | **94.69** | 90.83     | 53.30     | 55.00     | 58.83     | 72.27     |
>
> The result shows that **AdaR still yields substantial performance gains (+5.86) and effectively mitigates robustness failures that persist under test-time-scaling-enabled "Qwen3-4B-Thinking-2507" (+1.51)**.
> We additionally observe that the model's rollout time on seed dataset is shorter than that of AdaR ($\approx0.36\times$), suggesting that generating correct answers becomes more difficult when the model is exposed to mild numerical perturbations, revealing pronounced brittleness.
> Taken together, these findings highlight a pervasive limitation in existing models: **strong performance on conventional test sets often obscures fundamental weaknesses in generalization and robustness.**
> In contrast, AdaR effectively exposes and remedies such latent vulnerabilities, demonstrating its broad applicability.
> Additional details are provided in Appendix 6 in the updated PDF.
>
> *Owing to the inclusion of additional baselines and an unexpected computing resource failure, we regret that the experimental results on the strong baseline are delivered 2 days later than originally anticipated.*

---

### Official Review · Reviewer_9FAY · 2025-10-31

**Soundness:** 2
**Presentation:** 3
**Contribution:** 2
**Rating:** 4
**Confidence:** 3

**Summary:**

This paper proposes a refinement of Reinforcement Learning with Verifiable Rewards (RLVR, introduced in Lambert et al. 2025) for fine-tuning LLM to reduce spurious reasoning in mathematical problem solving. It introduces a framework for Adaptive Reasoning that aims to incentivize the LLM to provide a logic or program that can solve not only a given query but also similar queries with different variable parameters. The training data set for fine-tuning is created from a 9k subset of ORCA-Math, using an LLM to generate query templates and problem logics for supervised reinforcement learning. In contrast to RLVR, which rewards correct answers, AdaR also checks for the correct problem-solving logic. Qwen2.5-MATH, DeepSeekMath, and Llama3 are tested on GSM8K, ORCA-Adar-test (a 2.5k subset of ORCA-MATH without the 9k for AdaR training), GSM-SYM, MATH, CollegeMATH, CollegeMath, TheoremQA, and AIME, fine-tuned with AdaR and compared against supervised fine-tuning (SFT) with the negative log-likelihood objective, and two other methods, MetaMATH (Yu et al., 2023), and MathGenie (Lu et al., 2024). The paper reports small to moderate performance improvements with AdaR across all the different datasets.

**Strengths:**

Overall, the paper presents a solid approach to enhancing LLM fine-tuning for mathematical problem-solving. The summary of the method and experiments is well-structured and easy to follow. While the core idea of incentivizing LLM to mimic adaptive reasoning to improve CoT is not new, the implementation of adapting RLVR and synthesizing perturbed data for the fine-tuning appears to be well-executed.

**Weaknesses:**

1.  The main weakness is the lack of any proper discussion of the limitations of the proposed method. As fine-tuning on LLM-synthesized problem perturbations may improve LLM performance to some degree, the results indicate that there are still relevant limitations, either in adaptive reasoning in principle or in AdaR's ability to enforce such reasoning. In its current form, the Conclusion does not discuss the results at all, but rather provides a mere summary (which is completely redundant with the introduction). Besides a discussion of results, the paper could also include a slightly more theoretical discussion on what we can expect from such a method, attempting to get LLM to mimic adaptive reasoning.
2.  A second weakness could be the limited scope of the paper, mainly providing a fine-tuning approach with moderate results, but as I am no expert in LLM-finetuning, I am not sure about the expected scope of an ICLR paper in this area.

**Questions:**

Besides the addressed weaknesses of a missing discussion of results and limitations of the approach and method, the following could improve the clarity of the abstract:

1.  The abstract should be more precise and include that AdaR is an LLM fine-tuning approach.
2.  RLVR should be spelled out and properly referenced in the abstract.

Also note that Figure 1 is too small to be properly readable when printed out, and even at a reasonable zoom level.

---

> ### Author Response · Authors · 2025-11-27
>
> # [W1] Lack of discussion of limitations
>
> Our method has several limitations. First, while Section 4.3 highlights the importance of scaling $L$, the current AdaR implementation adjusts only $T$ and $x$, leaving $L$ as a future extension. Second, AdaR assumes that each query’s reasoning process can be represented as an executable program with perturbable inputs, which implies that applying AdaR in more general settings (e.g., theorem proving) would require more sophisticated designs. Third, owing to computational constraints, we do not evaluate AdaR with reasoning LLMs as base models. We have added the Limitation section to the updated PDF.
>
> Regarding more theoretical discussion, as stated in the *Overall Response*, our synthetic data methodology in AdaR is derived from the modeling principles underlying Adaptive Reasoning.
>
> # [W2] Limited scope of this paper
> Our method is not a fine-tuning approach, and the contributions have already been clearly enumerated in the *Overall Response*. Moreover, the effectiveness of our method is substantial. As highlighted in Observations 1 and 2 in Section 3.2, using only 9K synthetic samples based on primary-school math problems, AdaR achieves significant performance gains: a +4.75 improvement on the robustness test set and a +8.56 improvement on out-of-domain test sets with greater difficulty. In addition, the method demonstrates strong scalability: as the data size increases, performance continues to improve (Section 4.3).
>
> # [Q1] Suggestions for revisions to AdaR in the abstract
> As W1 noted, AdaR is not a fine-tuning approach.
>
> # [Q2] Suggestions for revisions to RLVR in the abstract
> We have added the content "Reinforcement Learning with Verifiable Rewards (RLVR)" into the abstract and cited it in the proper position of the Introduction.

---

### Author Response · Authors · 2025-11-27
**Overall Response**

We would like to sincerely thank all reviewers for your time and effort in evaluating our manuscript.  Below, we summarize the key contributions we would like to emphasize in this rebuttal, organized into three points.
# 1. Importance of the Adaptive Reasoning formulation
Based on prior work [1,2] and our preliminary experiments (Section 4.1), existing LLMs exhibit unstable performance when addressing robustness and generalization problems. For example, the query “Find A such that 32 × A × A × A = 42592” can be solved reliably by current models, whereas the seemingly analogous query “Find A such that 29 × A × A × A = 707281” often fails to yield the correct solution. We therefore propose the formulation of Adaptive Reasoning, which emphasizes algebraic thinking (the green arrow in Figure 1). In Adaptive Reasoning, a model is required to
- **Decompose** each query $q$ into an algebraically abstracted template $T$,
- **Record** the corresponding variable values as $x$,
- **Model** the underlying reasoning logic $L$ as a function over x conditioned on $T$,
- **Apply** x to the learned logic $L_T$ to produce the final answer $y.$

From a data-driven perspective, Adaptive Reasoning reveals that the three components $(T, L, x)$ can be independently perturbed to improve generalization. Prior work focuses primarily on only a subset of these dimensions, such as perturbing $T$ (i.e., paraphrasing in MetaMath [3]) or enriching $L$ (i.e., increasing data coverage), whereas our work additionally provides a systematic treatment of $x$.  The definition of **Adaptive Reasoning** is thus both meaningful and insightful: **it helps elucidate the fundamental mechanisms underlying robustness and generalization problems, and it highlights that AdaR considers all three components—$T$ (paraphrasing), $L$ (data incorporation), and $x$ (values perturbations)**, thereby yielding a more comprehensive training signal that better supports the modeling of Adaptive Reasoning.
# 2. Innovation of the AdaR training framework.
To operationalize Adaptive Reasoning, we propose AdaR, a training framework with two tightly coupled components:

**An automated synthesis pipeline with high quality.**  Besides paraphrasing, although "value perturbation" may appear conceptually straightforward, prior work has not realized high-fidelity, scalable perturbations. For example, GSM-Symbolic [1] relies on manually designed templates, while MathGenie [4] employs LLM-based perturbations that are inherently simple and weakly controlled (Section 5).  **AdaR is, to our knowledge, the first to automate this goal.** It introduces external code execution, controllable perturbations, and leverages cross-verification between programs and LLMs, thereby substantially improving the quality of the synthesized data.

**RL-based training method (RLVR) that uses this synthetic data.**  Furthermore, **the synergy between our synthetic data and RLVR is essential** (Section 2.2): When performing RLVR on the original training data, the model may produce correct answers simply by memorizing CoTs from SFT, even when the underlying reasoning is spurious. In contrast, our synthetic data makes such memorization much more likely to break, leading to more effective identification and penalization of spurious reasoning in RLVR.
# 3. Valuable analytical insights
  - **We introduce an analysis method that disentangles a model’s mathematical calculation ability from its reasoning ability.** Using this method, we identify the underlying cause of an important phenomenon: although SFT can improve benchmark accuracy, it contributes minimally to genuine generalization. Specifically, SFT tends to enhance a model’s calculation ability while simultaneously degrading its reasoning ability, yet reasoning ability is a critical determinant of a model's capacity to generalize (Section 3.3)
  - Recognized by reviewer Rq4c, **we introduce an innovative metric Influence of Logical Order (ILO)** that evaluates a model’s reasoning robustness by measuring the change in answer perplexity after shuffling the CoT sentence.
  - **We provide a unified perspective on prior mathematical data synthesis methods** by categorizing them along three dimensions, query-template q, problem-solving logic L, and values x, and evaluating their marginal contributions. Our results show that AdaR’s targeted enhancement of x is more effective than T. (Section 4.2)

*The corresponding revisions have been implemented, and all modifications are highlighted with a yellow background in the updated PDF.*

[1] Gsm-symbolic: Understanding the limitations of mathematical reasoning in large language models

[2] Evaluating mathematical reasoning across large language models: A fine-grained approach.

[3] Metamath: Bootstrap your own mathematical questions for large language models.

[4] Mathgenie: Generating synthetic data with question back-translation for enhancing mathematical reasoning of llms.

---

### Author Response · Authors · 2025-12-03
**Summary of Rebuttal for AC**

Based on prior work [1, 2] and our preliminary experiments (Section 4.1), existing LLMs exhibit unstable performance when addressing robustness and generalization deficiencies. For example, a model may correctly answer a query that it has likely seen before, such as "Find A such that 32 × A × A × A = 42592." But if we change the numbers, e.g., "Find A such that 29 × A × A × A = 707281", the model often fails to yield the correct solution.

## What causes these deficiencies?

To robustly generalize across diverse numerical values $x$, the model must approximate function $L(x)$. Acquiring such a function requires the model to correctly interpret the query and to disentangle the numerical values $x$ from the query itself, an ability that reflects algebraic reasoning. We therefore introduce the formulation of Adaptive Reasoning to express this process (the green arrow in Figure 1). It contains 3 essential components: **(1) the query template $T$**, **(2) the corresponding problem-solving logic $L$**, and **(3) the numerical values $x$**.

However, previous synthetic data approaches **predominantly focused on perturbing the template $T$** (e.g., paraphrasing) **while largely neglecting systematic perturbations of**  $x$. This omission is critical, as diversity in $x$ is necessary for models to approximate the underlying reasoning function $L_T$ and instead overfit to fixed $x$.

## How to solve?

Our method, AdaR, is **the first to jointly incorporate all three dimensions**  $T$ (paraphrasing), $L$ (data expansion), and $x$ (numerical perturbations). Crucially, AdaR introduces **high-precision, controllable numerical perturbations at scale, while ensuring verified gold answers** (recognized by Reviewers Rq4c, prZi, and k8Au).  Although value perturbations are trivial, producing accurate gold answers at the same time has proven non-trivial. AdaR is the first to achieve this reliably and at scale. We further clarify that these **synthetic data, when combined with RLVR,** effectively **mitigate the cases where ORM incorrectly rewards reasoning trajectories driven by memorization or guessing.**

## More Contributions?

Comprehensive experiments provide several notable contributions:

1. **Even the recent test-time scaling models (e.g., Qwen3-4B-Thinking-2507) still exhibit robustness deficiencies** on math problems, including those at primary-school level, while AdaR effectively mitigates these deficiencies and further strengthens their performance. (Section 4.5)
2. We introduce a technique that decouples mathematical performance into reasoning ability and calculation ability, enabling a clearer illustration of the benefits obtained by combining synthetic data with RLVR. (Section 3.3)
3. Although SFT can improve benchmark accuracy, it contributes minimally to genuine generalization. Specifically, we find that SFT tends to enhance a model's calculation ability while simultaneously degrading its reasoning ability, yet reasoning ability is a critical determinant of a model's capacity to generalize (Section 3.3)
4. We introduce a novel metric for evaluating reasoning robustness (recognized by Reviewer Rq4c in Section 4.1).

## Is this work complete?

Following discussions with the reviewers, we have made the paper more complete:

- Added experiments on stronger baselines (addressing Reviewers Rq4c and prZi)

- Expanded the discussion on *Limitations* (addressing Reviewer 9FAY)

- Provided additional analyses of the training dynamics; Further clarified important experimental settings; Attempted to revise the title for greater accuracy (addressing Reviewer k8Au)

We have responded explicitly and thoroughly to every weakness and question raised by each reviewer. We sincerely appreciate their feedback, which has significantly improved the quality and clarity of the paper.

[1] Gsm-symbolic: Understanding the limitations of mathematical reasoning in large language models

[2] Evaluating mathematical reasoning across large language models: A fine-grained approach.

---

### Note · Authors · 2026-01-06

I have read and agree with the venue's withdrawal policy on behalf of myself and my co-authors.